# Linear Multi-Resource Allocation with Semi-Bandit Feedback

**Tor Lattimore**
Department of Computing Science
University of Alberta, Canada
tor.lattimore@gmail.com

**Koby Crammer**
Department of Electrical Engineering
The Technion, Israel
koby@ee.technion.ac.il

**Csaba Szepesvári**
Department of Computing Science
University of Alberta, Canada
szepesva@ualberta.ca

## Abstract

We study an idealised sequential resource allocation problem. In each time step the learner chooses an allocation of several resource types between a number of tasks. Assigning more resources to a task increases the probability that it is completed. The problem is challenging because the alignment of the tasks to the resource types is unknown and the feedback is noisy. Our main contribution is the new setting and an algorithm with nearly-optimal regret analysis. Along the way we draw connections to the problem of minimising regret for stochastic linear bandits with heteroscedastic noise. We also present some new results for stochastic linear bandits on the hypercube that significantly improve on existing work, especially in the sparse case.

## 1 Introduction

Economist Thomas Sowell remarked that "The first lesson of economics is scarcity: There is never enough of anything to fully satisfy all those who want it."[1] The optimal allocation of resources is an enduring problem in economics, operations research and daily life. The problem is challenging not only because you are compelled to make difficult trade-offs, but also because the (expected) outcome of a particular allocation may be unknown and the feedback noisy.

We focus on an idealised resource allocation problem where the economist plays a repeated resource allocation game with multiple resource types and multiple tasks to which these resources can be assigned. Specifically, we consider a (nearly) linear model with $D$ resources and $K$ tasks. In each time step $t$ the economist chooses an allocation of resources $M_t \in \mathbb{R}^{D \times K}$ where $M_{tk} \in \mathbb{R}^D$ is the $k$th column and represents the amount of each resource type assigned to the $k$th task. We assume that the $k$th task is completed successfully with probability $\min\{1, \langle M_{tk}, \nu_k \rangle\}$ and $\nu_k \in \mathbb{R}^D$ is an unknown non-negative vector that determines how the success rate of a given task depends on the quantity and type of resources assigned to it. Naturally we will limit the availability of resources by demanding that $M_t$ satisfies $\sum_{k=1}^{K} M_{tdk} \leq 1$ for all resource types $d$. At the end of each time step the economist observes which tasks were successful. The objective is to maximise the number of successful tasks up to some time horizon $n$ that is known in advance. This model is a natural generalisation of the one used by Lattimore et al. [2014], where it was assumed that there was a single resource type only.

An example application might be the problem of allocating computing resources on a server between a number of Virtual Private Servers (VPS). In each time step (some fixed interval) the controller chooses how much memory/cpu/bandwidth to allocate to each VPS. A VPS is said to fail in a given round if it fails to respond to requests in a timely fashion. The requirements of each VPS are unknown in advance, but do not change greatly with time. The controller should learn which VPS benefit the most from which resource types and allocate accordingly.

The main contribution of this paper besides the new setting is an algorithm designed for this problem along with theoretical guarantees on its performance in terms of the regret. Along the way we present some additional results for the related problem of minimising regret for stochastic linear bandits on the hypercube. We also prove new concentration results for weighted least squares estimation, which may be independently interesting.

The generalisation of the work of Lattimore et al. [2014] to multiple resources turns out to be fairly non-trivial. Those with knowledge of the theory of stochastic linear bandits will recognise some similarity. In particular, once the nonlinearity of the objective is removed, the problem is equivalent to playing $K$ linear bandits in parallel, but where the limited resources constrain the actions of the learner and correspondingly the returns for each task. Stochastic linear bandits have recently been generating a significant body of research (e.g., Auer [2003], Dani et al. [2008], Rusmevichientong and Tsitsiklis [2010], Abbasi-Yadkori et al. [2011, 2012], Agrawal and Goyal [2012] and many others). A related problem is that of online combinatorial optimisation. This has an extensive literature, but most results are only applicable for discrete action sets, are in the adversarial setting, and cannot exploit the additional structure of our problem. Nevertheless, we refer the interested reader to (say) the recent work by Kveton et al. [2014] and references there-in. Also worth mentioning is that the resource allocation problem at hand is quite different to the "linear semi-bandit" proposed and analysed by Krishnamurthy et al. [2015] where the action set is also finite (the setting is different in many other ways besides).

Given its similarity, it is tempting to apply the techniques of linear bandits to our problem. When doing so, two main difficulties arise. The first is that our payoffs are non-linear: the expected reward is a linear function only up to a point after which it is clipped. In the resource allocation problem this has a natural interpretation, which is that over-allocating resources beyond a certain point is fruitless. Fortunately, one can avoid this difficulty rather easily by ensuring that with high probability resources are never over-allocated. The second problem concerns achieving good regret regardless of the task specifics. In particular, when the number of tasks $K$ is large and resources are at a premium the allocation problem behaves more like a $K$-armed bandit where the economist must choose the few tasks that can be completed successfully. For this kind of problem regret should scale in the worst case with $\sqrt{K}$ only [Auer et al., 2002, Bubeck and Cesa-Bianchi, 2012]. The standard linear bandits approach, on the other hand, would lead to a bound on the regret that depends linearly on $K$. To remedy this situation, we will exploit that if $K$ is large and resources are scarce, then many tasks will necessarily be under-resourced and will fail with high probability. Since the noise model is Bernoulli, the variance of the noise for these tasks is extremely low. By using weighted least-squares estimators we are able to exploit this and thereby obtain an improved regret. An added benefit is that when resources are plentiful, then all tasks will succeed with high probability under the optimal allocation, and in this case the variance is also low. This leads to a poly-logarithmic regret for the resource-laden case where the optimal allocation fully allocates every task.

## 2   Preliminaries

If $F$ is some event, then $\neg F$ is its complement (i.e., it is the event that $F$ does not occur). If $A$ is positive definite and $x$ is a vector, then $\|x\|_A^2 = x^\top A x$ stands for the weighted 2-norm. We write $|x|$ to be the vector of element-wise absolute values of $x$. We let $\nu \in \mathbb{R}^{D \times K}$ be a matrix with columns $\nu_1, \ldots \nu_K$. All entries in $\nu$ are non-negative, but otherwise we make no global assumptions on $\nu$. At each time step $t$ the learner chooses an allocation matrix $M_t \in \mathcal{M}$ where

$$\mathcal{M} = \left\{ M \in [0,1]^{D \times K} : \sum_{k=1}^{K} M_{dk} \leq 1 \text{ for all } d \right\}.$$

The assumption that each resource type has a bound of $1$ is non-restrictive, since the units of any resource can be changed to accommodate this assumption. We write $M_{tk} \in [0,1]^D$ for the $k$th

column of $M_t$. The reward at time step $t$ is $\|Y_t\|_1$ where $Y_{tk} \in \{0, 1\}$ is sampled from a Bernoulli distribution with parameter $\psi(\langle M_{tk}, \nu_k \rangle) = \min\{1, \langle M_{tk}, \nu_k \rangle\}$. The economist observes all $Y_{tk}$, however, not just the sum. The optimal allocation is denoted by $M^*$ and defined by

$$M^* = \arg\max_{M \in \mathcal{M}} \sum_{k=1}^{K} \psi(\langle M_k, \nu_k \rangle).$$

We are primarily concerned with designing an allocation algorithm that minimises the expected (pseudo) regret of this problem, which is defined by

$$R_n = n \sum_{k=1}^{K} \psi(\langle M_k^*, \nu_k \rangle) - \mathbb{E}\left[\sum_{t=1}^{n} \sum_{k=1}^{K} \psi(\langle M_{tk}, \nu_k \rangle)\right],$$

where the expectation is taken over both the actions of the algorithm and the observed reward.

**Optimal Allocations**

If $\nu$ is known, then the optimal allocation can be computed by constructing an appropriate linear program. Somewhat surprisingly it may also be computed exactly in $O(K \log K + D \log D)$ time using Algorithm 1 below. The optimal allocation is not so straight-forward as, e.g., simply allocating resources to the incomplete task for which the corresponding $\nu$ is largest in some dimension. For example, for $K = 2$ tasks and $d = 2$ resource types:

$$\nu = \begin{pmatrix} \nu_1 & \nu_2 \end{pmatrix} = \begin{pmatrix} 0 & 1/2 \\ 1/2 & 1 \end{pmatrix} \implies M^* = \begin{pmatrix} M_1^* & M_2^* \end{pmatrix} = \begin{pmatrix} 0 & 1 \\ 1/2 & 1/2 \end{pmatrix}.$$

We see that even though $\nu_{22}$ is the largest parameter, the optimal allocation assigns only half of the second resource ($d = 2$) to this task. The right approach is to allocate resources to incomplete tasks using the ratios as prescribed by Algorithm 1. The intuition for allocating in this way is that resources should be allocated as efficiently as possible, and efficiency is determined by the ratio of the expected success due to the allocation of a resource and the amount of resources allocated.

**Theorem 1.** *Algorithm 1 returns $M^*$.*

---

**Algorithm 1**

---

**Input:** $\nu$
$M = 0 \in \mathbb{R}^{D \times K}$ and $B = \mathbb{1} \in \mathbb{R}^D$
**while** $\exists\, k, d$ s.t $\langle M_k, \nu_k \rangle < 1$ and $B_d > 0$ **do**

  $\mathcal{A} = \{k : \langle M_k, \nu_k \rangle < 1\}$ and $\mathcal{B} = \{d : B_d > 0\}$

  $k, d = \arg\max_{(k,d) \in \mathcal{A} \times \mathcal{B}} \min_{i \in \mathcal{A} \setminus \{k\}} \left( \frac{\nu_{dk}}{\nu_{di}} \right)$

  $M_{dk} = \min\left\{ B_d, \frac{1 - \langle M_k, \nu_k \rangle}{\nu_{dk}} \right\}$

**end while**
**return** $M$

---

The proof of Theorem 1 and an implementation of Algorithm 1 may be found in the supplementary material.

We are interested primarily in the case when $\nu$ is unknown, so Algorithm 1 will not be directly applicable. Nevertheless, the algorithm is useful as a module in the implementation of a subsequent algorithm that estimates $\nu$ from data.

## 3 Optimistic Allocation Algorithm

We follow the optimism in the face of uncertainty principle. In each time step $t$, the algorithm constructs an estimator $\hat{\nu}_{kt}$ for each $\nu_k$ and a corresponding confidence set $C_{tk}$ for which $\nu_k \in C_{tk}$ holds with high probability. The algorithm then takes the optimistic action subject to the assumption that $\nu_k$ does indeed lie in $C_{tk}$ for all $k$. The main difficulty is the construction of the confidence sets. Like other authors [Dani et al., 2008, Rusmevichientong and Tsitsiklis, 2010, Abbasi-Yadkori et al., 2011] we define our confidence sets to be ellipses, but the use of a weighted least-squares estimator means that our ellipses may be significantly smaller than the sets that would be available by using these previous works in a straightforward way. The algorithm accepts as input the number of tasks and resource types, the horizon and constants $\alpha > 0$ and $\beta$ where constant $\beta$ is defined by

$$\delta = \frac{1}{nK}, \qquad N = \left(4n^4 D^2\right)^D, \qquad B \geq \max_k \|\nu_k\|_2^2, \qquad \text{so that}$$

$$\beta = \left(1 + \sqrt{\alpha B} + 2\sqrt{\log\left(\frac{6nN}{\delta} \log\left(\frac{3nN}{\delta}\right)\right)}\right)^2. \tag{1}$$

Note that $B$ must be a known bound on $\max_k \|\nu_k\|_2^2$, which might seem like a serious restriction, until one realizes that it is easy to add an initialisation phase where estimates are quickly made while incurring minimal additional regret, as was also done by Lattimore et al. [2014]. The value of $\alpha$ determines the level of regularisation in the least squares estimation and will be tuned later to optimise the regret.

---

**Algorithm 2** Optimistic Allocation Algorithm

---

1: **Input** $K$, $D$, $n$, $\alpha$, $\beta$
2: **for** $t \in 1, \ldots, n$ **do**
3:       // **Compute confidence sets for all tasks** $k$**:**
4:       $G_{tk} = \alpha I + \sum_{\tau < t} \gamma_{\tau k} M_{\tau k} M_{\tau k}^\top$
5:       $\hat{\nu}_{tk} = G_{tk}^{-1} \sum_{\tau < t} \gamma_{\tau k} M_\tau Y_{\tau k}$
6:       $C_{tk} = \left\{ \tilde{\nu}_k : \|\tilde{\nu}_k - \hat{\nu}_{tk}\|_{G_{tk}}^2 \leq \beta \right\}$ and $C'_{tk} = \left\{ \tilde{\nu}_k : \|\tilde{\nu}_k - \hat{\nu}_{tk}\|_{G_{tk}}^2 \leq 4\beta \right\}$
7:       // **Compute optimistic allocation:**
8:       $M_t = \arg\max_{M_t \in \mathcal{M}} \max_{\tilde{\nu}_k \in C_{tk}} \psi(\langle M_{tk}, \tilde{\nu}_k \rangle)$
9:       // **Observe success indicators** $Y_{tk}$ **for all tasks** $k$**:**
10:      $Y_{tk} \sim \text{Bernoulli}(\psi(\langle M_{tk}, \nu_k \rangle))$
11:      // **Compute weights for all tasks** $k$**:**
12:      $\gamma_{tk}^{-1} = \arg\max_{\tilde{\nu}_k \in C'_{tk}} \langle M_{tk}, \tilde{\nu}_k \rangle (1 - \langle M_{tk}, \tilde{\nu}_k \rangle)$
13: **end for**

---

**Computational Efficiency**

We could not find an efficient implementation of Algorithm 2 because solving the bilinear optimisation problem in Line 8 is likely to be NP-hard (Bennett and Mangasarian [1993] and also Petrik and Zilberstein [2011]). In our experiments we used a simple algorithm based on optimising for $M$ and $\nu$ in alternative steps combined with random restarts, but for large $D$ and $K$ this would likely not be efficient. In the supplementary material we present an alternative algorithm that is efficient, but relies on the assumption that $\|\nu_k\|_1 \leq 1$ for all $k$. In this regime it is impossible to over-allocate resources and this fact can be exploited to obtain an efficient and practical algorithm with strong guarantees. Along the way, we are able to construct an elegant algorithm for linear bandits on the hypercube that enjoys optimal regret and adapts to sparsity.

Computing the weights $\gamma_{tk}$ (Line 12) is (somewhat surprisingly) straight-forward. Define

$$\bar{p}_{tk} = \langle M_{tk}, \hat{\nu}_{tk} \rangle + 2\sqrt{\beta} \|M_{tk}\|_{G_{tk}^{-1}} \quad \text{and} \quad \underline{p}_{tk} = \langle M_{tk}, \hat{\nu}_{tk} \rangle - 2\sqrt{\beta} \|M_{tk}\|_{G_{tk}^{-1}} \, .$$

Then the weights can be computed by

$$\gamma_{tk}^{-1} = \begin{cases} \bar{p}_{tk}(1 - \bar{p}_{tk}) & \text{if } \bar{p}_{tk} \leq \frac{1}{2} \\ \underline{p}_{tk}(1 - \underline{p}_{tk}) & \text{if } \underline{p}_{tk} \geq \frac{1}{2} \\ \frac{1}{4} & \text{otherwise} \, . \end{cases} \tag{2}$$

A curious reader might wonder why the weights are computed by optimising within confidence set $C'_{tk}$, which has double the radius of $C_{tk}$. The reason is rather technical, but essentially if the true parameter $\nu_k$ were to lie on the boundary of the confidence set, then the corresponding weight could become infinite. For the analysis to work we rely on controlling the size of the weights. It is not clear whether or not this trick is really necessary.

## 4   Worst-case Regret for Algorithm 2

We now analyse the regret of Algorithm 2. First we offer a worst-case bound on the regret that depends on the time-horizon like $O(\sqrt{n})$. We then turn our attention to the resource-laden case where the optimal allocation satisfies $\langle M_k^*, \nu_k \rangle = 1$ for all $k$. In this instance we show that the dependence on the horizon is only poly-logarithmic, which would normally be unexpected when the

action-space is continuous. The improvement comes from the weighted estimation that exploits the fact that the variance of the noise under the optimal allocation vanishes.

**Theorem 2.** *Suppose Algorithm 2 is run with bound* $B \geq \max_k \|\nu_k\|_2^2$. *Then*

$$R_n \leq 1 + 4D \sqrt{2\beta nK \left( \max_k \|\nu_k\|_\infty + 4\sqrt{\beta/\alpha} \right) \log(1 + 4n^2)}.$$

Choosing $\alpha = B^{-1} \log \left( \frac{6nN}{\delta} \log \left( \frac{3nN}{\delta} \right) \right)$ and assuming that $B \in O(\max_k \|\nu_k\|_2^2)$, then

$$R_n \in O \left( D^{3/2} \sqrt{nK \max_k \|\nu_k\|_2 \log n} \right).$$

The proof of Theorem 2 will follow by carefully analysing the width of the confidence sets as the algorithm makes allocations. We start by proving the validity of the confidence sets, and then prove the theorem.

### Weighted Least Squares Estimation

For this sub-section we focus on the problem of estimating a single unknown $\nu = \nu_k$. Let $M_1, \ldots, M_n$ be a sequence of allocations to task $k$ with $M_t \in \mathbb{R}^D$. Let $\{\mathcal{F}_t\}_{t=0}^n$ be a filtration with $\mathcal{F}_t$ containing information available at the end of round $t$, which means that $M_t$ is $\mathcal{F}_{t-1}$-measurable. Let $\gamma_1, \ldots, \gamma_n$ be the sequence of weights chosen by Algorithm 2. The sequence of outcomes is $Y_1, \ldots, Y_n \in \{0, 1\}$ for which $\mathbb{E}[Y_t|\mathcal{F}_{t-1}] = \psi(\langle M_t, \nu \rangle)$. The weighted regularised gram matrix is $G_t = \alpha I + \sum_{\tau < t} \gamma_\tau M_\tau M_\tau^\top$ and the corresponding weighted least squares estimator is

$$\hat{\nu}_t = G_t^{-1} \sum_{\tau < t} \gamma_t M_\tau Y_\tau.$$

**Theorem 3.** *If* $\|\nu\|_2^2 \leq B$ *and* $\beta$ *is chosen as in Eq. (1), then* $\|\hat{\nu}_t - \nu\|_{G_t}^2 \leq \beta$ *for all* $t \leq n$ *with probability at least* $1 - \delta = 1/(nK)$.

Similar results exist in the literature for unweighted least-squares estimators (for example, Dani et al. [2008], Rusmevichientong and Tsitsiklis [2010], Abbasi-Yadkori et al. [2011]). In our case, however, $G_t$ is the weighted gram matrix, which may be significantly larger than an unweighted version when the weights become large. The proof of Theorem 3 is unfortunately too long to include in the main text, but it may be found in the supplementary material.

### Analysing the Regret

We start with some technical lemmas. Let $F$ be the failure event that $\|\hat{\nu}_{tk} - \nu_k\|_{G_{tk}}^2 > \beta$ for some $t \leq n$ and $1 \leq k \leq K$.

**Lemma 4** (Abbasi-Yadkori et al. [2012]). *Let* $x_1, \ldots, x_n$ *be an arbitrary sequence of vectors with* $\|x_t\|_2^2 \leq c$ *and let* $G_t = I + \sum_{s=1}^{t-1} x_s x_s^\top$. *Then* $\sum_{t=1}^n \min \left\{ 1, \|x_t\|_{G_t^{-1}}^2 \right\} \leq 2D \log \left( 1 + \frac{c \cdot n}{D} \right)$.

**Corollary 5.** *If* $F$ *does not hold, then* $\sum_{t=1}^n \gamma_{tk} \min \left\{ 1, \|M_{tk}\|_{G_{tk}^{-1}}^2 \right\} \leq 8D \log(1 + 4n^2)$.

The proof is omitted, but follows rather easily by showing that $\gamma_{tk}$ can be moved inside the minimum at a price of increasing the loss at most by a factor of four, and then applying Lemma 4. See the supplementary material for the formal proof.

**Lemma 6.** *Suppose* $F$ *does not hold, then* $\sum_{k=1}^K \gamma_{tk}^{-1} \leq D \left( \max_k \|\nu_k\|_\infty + 4\sqrt{\beta/\alpha} \right)$.

*Proof.* We exploit the fact that $\gamma_{tk}^{-1}$ is an estimate of the variance, which is small whenever $\|M_{tk}\|_1$ is small:

$$\gamma_{tk}^{-1} = \arg\max_{\tilde{\nu}_k \in C'_{tk}} \langle M_{tk}, \tilde{\nu}_k \rangle \left(1 - \langle M_{tk}, \tilde{\nu}_k \rangle\right) \leq \arg\max_{\tilde{\nu}_k \in C'_{tk}} \langle M_{tk}, \tilde{\nu}_k \rangle$$

$$= \langle M_{tk}, \nu \rangle + \arg\max_{\tilde{\nu}_k \in C_{tk'}} \langle M_{tk}, \tilde{\nu}_k - \nu \rangle \overset{(a)}{\leq} \|M_{tk}\|_1 \|\nu_k\|_\infty + 4\sqrt{\beta} \|M_{tk}\|_{G_{tk}^{-1}}$$

$$\overset{(b)}{\leq} \|M_{tk}\|_1 \|\nu_k\|_\infty + 4\sqrt{\beta} \|M_{tk}\|_{I/\alpha} \overset{(c)}{\leq} \|M_{tk}\|_1 \left(\|\nu_k\|_\infty + 4\sqrt{\beta/\alpha}\right),$$

where (a) follows from Cauchy-Schwartz and the fact that $\nu_k \in C'_{tk}$, (b) since $G_{tk}^{-1} \leq I/\alpha$ and basic linear algebra, (c) since $\|M_{tk}\|_{I/\alpha} = \sqrt{1/\alpha}\|M_{tk}\|_2 \leq \sqrt{1/\alpha}\|M_{tk}\|_1$. The result is completed since the resource constraints implies that $\sum_{k=1}^K \|M_{tk}\|_1 \leq D$. $\qquad\square$

*Proof of Theorem 2.* By Theorem 3 we have that $F$ holds with probability at most $\delta = 1/(nK)$. If $F$ does not hold, then by the definition of the confidence set we have $\nu_k \in C_{tk}$ for all $t$ and $k$. Therefore

$$R_n = \mathbb{E}\sum_{t=1}^n \sum_{k=1}^K \left(\langle M_k^*, \nu_k \rangle - \psi(\langle M_{tk}, \nu_k \rangle)\right) \leq 1 + \mathbb{E}\left[\mathbb{1}\left\{\neg F\right\} \sum_{t=1}^n \sum_{k=1}^K \langle M_k^* - M_{tk}, \nu_k \rangle\right].$$

Note that we were able to replace $\psi(\langle M_{tk}, \nu_k \rangle) = \langle M_{tk}, \nu_k \rangle$, since if $F$ does not hold, then $M_{tk}$ will never be chosen in such a way that resources are over-allocated. We will now assume that $F$ does not hold and bound the argument in the expectation. By the optimism principle we have:

$$\sum_{t=1}^n \sum_{k=1}^K \langle M_k^* - M_{tk}, \nu_k \rangle \overset{(a)}{\leq} \sum_{t=1}^n \sum_{k=1}^K \min\left\{1, \langle M_{tk}, \tilde{\nu}_{tk} - \nu_k \rangle\right\}$$

$$\overset{(b)}{\leq} \sum_{t=1}^n \sum_{k=1}^K \min\left\{1, \|M_{tk}\|_{G_{tk}^{-1}} \|\tilde{\nu}_{tk} - \nu_k\|_{G_{tk}}\right\}$$

$$\overset{(c)}{\leq} 2\sum_{t=1}^n \sum_{k=1}^K \min\left\{1, \|M_{tk}\|_{G_{tk}^{-1}} \sqrt{\beta}\right\}$$

$$\overset{(d)}{\leq} 2\sqrt{n\sum_{t=1}^n \beta \left(\sum_{k=1}^K \min\left\{1, \|M_{tk}\|_{G_{tk}^{-1}}\right\}\right)^2}$$

$$\overset{(e)}{\leq} 2\sqrt{n\sum_{t=1}^n \beta \left(\sum_{k=1}^K \gamma_{tk}^{-1}\right) \left(\sum_{k=1}^K \gamma_{tk} \min\left\{1, \|M_{tk}\|_{G_{tk}^{-1}}^2\right\}\right)}$$

$$\overset{(f)}{\leq} 2\sqrt{nD\left(\max_k \|\nu_k\|_\infty + 4\sqrt{\frac{\beta}{\alpha}}\right) \sum_{t=1}^n \beta \left(\sum_{k=1}^K \gamma_{tk} \min\left\{1, \|M_{tk}\|_{G_{tk}^{-1}}^2\right\}\right)}$$

$$\overset{(g)}{\leq} 4D\sqrt{2\beta nK\left(\max_k \|\nu_k\|_\infty + 4\sqrt{\frac{\beta}{\alpha}}\right)\log(1 + 4n^2)}.$$

where (a) follows from the assumption that $\nu_k \in C_{tk}$ for all $t$ and $k$ and since $M_t$ is chosen optimistically, (b) by the Cauchy-Schwarz inequality, (c) by the definition of $\tilde{\nu}_{kt}$, which lies inside $C_{tk}$, (d) by Jensen's inequality, (e) by Cauchy-Schwarz again, (f) follows from Lemma 6. Finally (g) follows from Corollary 5. $\qquad\square$

## 5  Regret in Resource-Laden Case

We now show that if there are enough resources such that the optimal strategy can complete every task with certainty, then the regret of Algorithm 2 is poly-logarithmic (in contrast to $O(\sqrt{n})$ otherwise). As before we exploit the low variance, but now the variance is small because $\langle M_{tk}, \nu_k \rangle$ is

close to 1, while in the previous section we argued that this could not happen too often (there is no contradiction as the quantity $\max_k \|\nu_k\|$ appeared in the previous bound).

**Theorem 7.** *If $\sum_{k=1}^{K} \langle M_k^*, \nu_k \rangle = K$, then $R_n \leq 1 + 8\beta K D \log(1 + 4n^2)$.*

*Proof.* We start by showing that the weights are large:

$$\gamma_{tk}^{-1} = \max_{\underline{\nu} \in C'_{tk}} \langle M_{tk}, \underline{\nu} \rangle \left(1 - \langle M_{tk}, \underline{\nu} \rangle\right) \leq \max_{\underline{\nu} \in C'_{tk}} \left(1 - \langle M_{tk}, \underline{\nu} \rangle\right)$$

$$\leq \max_{\bar{\nu}, \underline{\nu} \in C'_{tk}} \langle M_{tk}, \bar{\nu} - \underline{\nu} \rangle \leq \|M_{tk}\|_{G_{tk}^{-1}} \max_{\bar{\nu}, \underline{\nu} \in C'_{tk}} \|\bar{\nu} - \underline{\nu}\|_{G_{tk}} \leq \|M_{tk}\|_{G_{tk}^{-1}} 4\sqrt{\beta}\,.$$

Applying the optimism principle and using the bound above combined with Corollary 5 gives the result:

$$\mathbb{E} R_n \leq 1 + \mathbb{E}\left[\mathbb{1}\left\{\neg F\right\} \sum_{t=1}^{n} \sum_{k=1}^{K} \min\left\{1, \langle M_{tk}, \tilde{\nu}_{kt} - \nu_k \rangle\right\}\right]$$

$$\leq 1 + 2\mathbb{E}\left[\mathbb{1}\left\{\neg F\right\} \sum_{t=1}^{n} \sum_{k=1}^{K} \min\left\{1, \|M_{tk}\|_{G_{tk}^{-1}} \sqrt{\beta}\right\}\right]$$

$$= 1 + 2\mathbb{E}\left[\mathbb{1}\left\{\neg F\right\} \sum_{t=1}^{n} \sum_{k=1}^{K} \min\left\{1, \gamma_{tk}^{-1} \gamma_{tk} \|M_{tk}\|_{G_{tk}^{-1}}\right\} \sqrt{\beta}\right]$$

$$\leq 1 + 8\beta \, \mathbb{E}\left[\mathbb{1}\left\{\neg F\right\} \sum_{t=1}^{n} \sum_{k=1}^{K} \min\left\{1, \gamma_{tk} \|M_{tk}\|_{G_{tk}^{-1}}^2\right\}\right]$$

$$\leq 1 + 8\beta K D \log(1 + 4n^2)\,. \qquad \square$$

## 6   Experiments

We present two experiments to demonstrate the behaviour of Algorithm 2. All code and data is available in the supplementary material. Error bars indicate 95% confidence intervals, but sometimes they are too small to see (the algorithm is quite conservative, so the variance is very low). We used $B = 10$ for all experiments. The first experiment demonstrates the improvements obtained by using a weighted estimator over an unweighted one, and also serves to give some idea of the rate of learning. For this experiment we used $D = K = 2$ and $n = 10^6$ and

$$\nu = \begin{pmatrix} \nu_1 & \nu_2 \end{pmatrix} = \begin{pmatrix} 8/10 & 2/10 \\ 4/10 & 2 \end{pmatrix} \implies M^* = \begin{pmatrix} 1 & 0 \\ 1/2 & 1/2 \end{pmatrix} \quad \text{and} \quad \sum_{k=1}^{K} \langle M_k^*, \nu_k \rangle = 2\,,$$

where the $k$th column is the parameter/allocation for the $k$th task. We ran two versions of the algorithm. The first, exactly as given in Algorithm 2 and the second identical except that the weights were fixed to $\gamma_{tk} = 4$ for all $t$ and $k$ (this value is chosen because it corresponds to the minimum inverse variance for a Bernoulli variable). The data was produced by taking the average regret over 8 runs. The results are given in Fig. 1. In Fig. 2 we plot $\gamma_{tk}$. The results show that $\gamma_{tk}$ is increasing linearly with $t$. This is congruent with what we might expect because in this regime the estimation error should drop with $O(1/t)$ and the estimated variance is proportional to the estimation error. Note that the estimation error for the algorithm with $\gamma_{tk} = 4$ will be $O(\sqrt{1/t})$.

For the second experiment we show the algorithm adapting to the environment. We fix $n = 5 \times 10^5$ and $D = K = 2$. For $\alpha \in (0, 1)$ we define

$$\nu_\alpha = \begin{pmatrix} 1/2 & \alpha/2 \\ 1/2 & \alpha/2 \end{pmatrix} \implies M^* = \begin{pmatrix} 1 & 0 \\ 1 & 0 \end{pmatrix} \quad \text{and} \quad \sum_{k=1}^{K} \langle M_k^*, \nu_k \rangle = 1\,.$$

The unusual profile of the regret as $\alpha$ varies can be attributed to two factors. First, if $\alpha$ is small then the algorithm quickly identifies that resources should be allocated first to the first task. However, in the early stages of learning the algorithm is conservative in allocating to the first task to avoid over-allocation. Since the remaining resources are given to the second task, the regret is larger for small

$\alpha$ because the gain from allocating to the second task is small. On the other hand, if $\alpha$ is close to $1$, then the algorithm suffers the opposite problem. Namely, it cannot identify which task the resources should be assigned to. Of course, if $\alpha = 1$, then the algorithm must simply learn that all resources can be allocated safely and so the regret is smallest here. An important point is that the algorithm never allocates all its resources at the start of the process because this risks over-allocation, so even in "easy" problems the regret will not vanish.

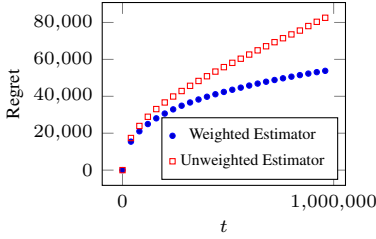

**Figure 1:** Weighted vs unweighted estimation

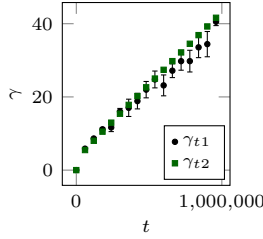

**Figure 2:** Weights

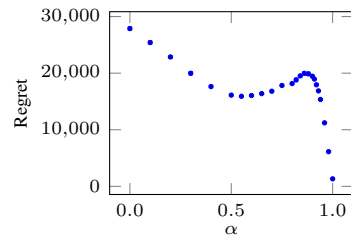

**Figure 3:** "Gap" dependence

## 7 Conclusions and Summary

We introduced the stochastic multi-resource allocation problem and developed a new algorithm that enjoys near-optimal worst-case regret. The main drawback of the new algorithm is that its computation time is exponential in the dimension parameters, which makes practical implementations challenging unless both $K$ and $D$ are relatively small. Despite this challenge we were able to implement that algorithm using a relatively brutish approach to solving the optimisation problem, and this was sufficient to present experimental results on synthetic data showing that the algorithm is behaving as the theory predicts, and that the use of the weighted least-squares estimation is leading to a real improvement.

Despite the computational issues, we think this is a reasonable first step towards a more practical algorithm as well as a solid theoretical understanding of the structure of the problem. As a consolation (and on their own merits) we include some other results:

- An efficient (both in terms of regret and computation) algorithm for the case where over-allocation is impossible.
- An algorithm for linear bandits on the hypercube that enjoys optimal regret bounds *and* adapts to sparsity.
- Theoretical analysis of weighted least-squares estimators, which may have other applications (e.g., linear bandits with heteroscedastic noise).

There are many directions for future research. The most natural is to improve the practicality of the algorithm. We envisage such an algorithm might be obtained by following the program below:

- Generalise the Thompson sampling analysis for linear bandits by Agrawal and Goyal [2012]. This is a highly non-trivial step, since it is no longer straight-forward to show that such an algorithm is optimistic with high probability. Instead it will be necessary to make do with some kind of local optimism for each task.
- The method of estimation depends heavily on the algorithm over-allocating its resources only with extremely low probability, but this significantly slows learning in the initial phases when the confidence sets are large and the algorithm is acting conservatively. Ideally we would use a method of estimation that depended on the real structure of the problem, but existing techniques that might lead to theoretical guarantees (e.g., empirical process theory) do not seem promising if small constants are expected.

It is not hard to think up extensions or modifications to the setting. For example, it would be interesting to look at an adversarial setting (even defining it is not so easy), or move towards a non-parametric model for the likelihood of success given an allocation.

## Footnotes

[1] He went on to add that "The first lesson of politics is to disregard the first lesson of economics." Sowell [1993]

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
