[Supplementary Material]

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

## A  Linear Algebra

We collect some well-known results in linear algebra for easy of reference. A square matrix $A$ is said to be positive definite (semi-definite) if it is symmetric and all its eigenvalues are positive (nonnegative). For $A$, $B$ positive definite, $A \leq B$ means $B - A$ is positive semi-definite.

**Proposition 8.** *Let $A$ and $B$ be positive-definite and $x$ and $y$ be vectors. The following hold:*

1. *If $A \leq B$, then $\|x\|_A \leq \|x\|_B$.*
2. *If $A \leq B$, then $A^{-1} \geq B^{-1}$.*
3. *If $A$ has maximum eigenvalue $\lambda_{\max}$, then $\|Ax\|_2 \leq \lambda_{\max} \|x\|_2$ and $\lambda_{\max} \leq \operatorname{trace}(A)$.*

# B  Concentration Bounds

**Theorem 9.** *Let $\delta \in (0, 1)$ and $X_1, \ldots, X_n$ be a sequence of random variables adapted to filtration $\{\mathcal{F}_t\}$ with $\mathbb{E}[X_t|\mathcal{F}_{t-1}] = 0$. Let $Z \subseteq \{1, \ldots, n\}$ be such that $\mathbb{1}\{t \in Z\}$ is $\mathcal{F}_{t-1}$-measurable and let $R_t$ be $\mathcal{F}_{t-1}$-measurable such that $|X_t| \leq R_t$ almost surely. Now define*

$$V = \sum_{t \in Z} \mathbb{V}[X_t|\mathcal{F}_{t-1}] + \sum_{t \notin Z} R_t^2/2\,, \qquad R = \max_{t \in Z} R_t\,, \qquad and \qquad S = \sum_{t=1}^n X_t\,.$$

*Then $\mathbb{P}\{S \geq f(R, V)\} \leq \delta$, where*

$$f(r, v) = \frac{2(r+1)}{3} \log \frac{2}{\delta_{r,v}} + \sqrt{2(v+1) \log \frac{2}{\delta_{r,v}}}\,, \qquad and$$

$$\delta_{r,v} = \frac{\delta}{3(r+1)^2(v+1)^2}\,.$$

The proof follows along precisely the same lines as the proof of Theorem 13 by Lattimore et al. [2014b], which itself is essentially just a modification of the Freedman's version of the Bernstein's inequality [Bernstein, 1946, Freedman, 1975]. The only modification required is to merge the proofs of Theorems 3.14 and 3.15 by McDiarmid [1998] using either Lemma 2.6 or 2.7 in that work depending on whether $t \in Z$ or otherwise. The intuition is that we are locally able to use either Hoeffding's lemma (Lemma 2.6) or Bennet's variance-dependent lemma (Lemma 2.7). See also the classical works by Bennett [1962] and Hoeffding [1963]. Once this is done, a simple peeling argument, identical to that used in the proof of their Theorem 13 by Lattimore et al. [2014b], is used.

# C  Proof of Theorem 3

Our approach generalises that used by Lattimore et al. [2014a] to the multi-dimensional case. Note that similar results were given by Dani et al. [2008], Rusmevichientong and Tsitsiklis [2010] and Abbasi-Yadkori et al. [2011], but none are able to effectively handle the heteroscedastic noise and so are unsuitable for our needs. Unfortunately we were not able to generalise the beautiful method of Abbasi-Yadkori et al. [2011], but our approach still enjoys relatively small constants and (in our view) is relatively insightful.

We will abbreviate the notation for simplicity. Pick some task $k$ and let $M_1, \ldots, M_t$ be a sequence of allocations chosen for it, $Y_1, \ldots, Y_t$ the corresponding rewards and $\gamma_1, \ldots, \gamma_t$ the weights as chosen by Algorithm 2. Fixing $k$, we omit the $k$-dependence in this section.

Recall that for $t \geq 1$ the gram matrix and weighted least-squares estimator are defined by

$$G_t = \alpha I + \sum_{s < t} \gamma_s M_s M_s^\top\,,$$

$$\hat{\nu}_t = G_t^{-1} \sum_{s < t} \gamma_s M_s Y_s\,.$$

We also set $G_0 = I$. Remember also that $Y_s$ is sampled from a Bernoulli distribution with parameter $\langle M_s, \nu \rangle$. Assuming that $\langle M_s, \nu \rangle \leq 1$, we can separate signal and noise by writing

$$Y_t = \langle M_s, \nu \rangle + \eta_s\,,$$

where $\eta_s \in [-1, 1]$, $\mathbb{E}[\eta_s|\mathcal{F}_{s-1}] = 0$ and $\mathbb{V}[\eta_s|\mathcal{F}_{s-1}] = \langle M_s, \nu \rangle (1 - \langle M_s, \nu \rangle)$. Note that the algorithm is crafted in such a way that $\langle M_s, \nu \rangle \leq 1$ unless some confidence interval fails, which only occurs on a low probability failure event $F_t$ to be defined shortly. For this reason we are able to ignore the non-linear part of the pay-off for this section, but the price we pay is that the confidence intervals must be chosen wide enough that the failure probability is very low, while other algorithms (such as UCB) are able to recover from failing confidence intervals. The confidence sets are given by

$$C_s = \left\{ \tilde{\nu} : \|\nu - \hat{\nu}_s\|_{G_s}^2 \leq \beta \right\} \qquad and \qquad C_s' = \left\{ \tilde{\nu} : \|\nu - \hat{\nu}_s\|_{G_s}^2 \leq 4\beta \right\}\,.$$

The key in the proof of Theorem 3 will be controlling the size of $S_t$ defined by

$$S_t = \sum_{s<t} \gamma_s \eta_s M_s \,.$$

Let $F_0 \subseteq F_1 \subseteq \cdots \subseteq F_n$ be a sequence of failure events defined by

$$F_t = \left\{ \exists s \leq t \text{ such that } \|S_s\|_{G_s^{-1}} + \sqrt{\alpha B} \geq \sqrt{\beta} \right\} \,.$$

**Lemma 10.** *Let $t \geq 1$. If $F_t$ does not hold, then $\nu \in C_t$.*

*Proof.* Since $F_t$ does not hold, we have that $\|S_t\|_{G_t^{-1}} + \sqrt{\alpha B} \leq \sqrt{\beta}$. Then,

$$
\begin{aligned}
\|\hat{\nu}_t - \nu\|_{G_t} &\overset{(a)}{=} \left\| G_t^{-1} S_t + G_t^{-1} \sum_{s<t} \gamma_s M_s M_s^\top \nu - G_t^{-1} G_t \nu \right\|_{G_t} \\
&\overset{(b)}{=} \left\| G_t^{-1} S_t - \alpha G_t^{-1} \nu \right\|_{G_t} \\
&\overset{(c)}{\leq} \|S_t\|_{G_t^{-1}} + \alpha \|\nu\|_{I/\alpha} \\
&\overset{(d)}{\leq} \sqrt{\beta} - \sqrt{\alpha B} + \sqrt{\alpha} \|\nu\|_2 \\
&\overset{(e)}{\leq} \sqrt{\beta} \,,
\end{aligned}
$$

where (a), (b) are immediate by substituting the definitions, (c) from the triangle inequality and Proposition 8.§2. Finally, (d) and (e) follow from the assumptions and because $\|\nu\|_2 \leq \sqrt{B}$. Therefore $\nu \in C_t$. $\qquad \square$

*Proof of Theorem 3.* Note that by definition $\beta \geq B$, hence $F_0$ holds. Let $t \leq n$ and assume that $F_{t-1}$ holds, which by Lemma 10 implies that $\nu \in C_s$ for all $s < t$. Shortly we will show that

$$\mathbb{P}\left\{ \|S_t\|_{G_t^{-1}} + \sqrt{\alpha B} \geq \sqrt{\beta} \text{ and not } F_{t-1} \right\} \leq \delta/n \,. \tag{3}$$

Then by induction we see that $\mathbb{P}\left\{ \neg F_n \right\} \geq 1 - \delta$, and so by Lemma 10 it follows that $\nu \in C_t$ for all $t \leq n$ with probability at least $1 - \delta$.

We now work on showing Eq. (3). Let $\lambda \in \mathbb{R}^D$ and define

$$
V_{s,\lambda} = \begin{cases} \mathbb{V}[\eta_s | \mathcal{F}_{s-1}] \gamma_s^2 \langle M_s, \lambda \rangle^2 \,, & \text{if } \gamma_s > 4; \\ \gamma_s \langle M_s, \lambda \rangle^2 \,, & \text{otherwise,} \end{cases}
$$

$$R_\lambda = \max_{s<t} \left\{ \gamma_s \langle M_s, \lambda \rangle : \gamma_s > 4 \right\} \,.$$

Then, by Theorem 9 we have with probability at least $1 - \delta/n$ that

$$
\begin{aligned}
\langle S_t, \lambda \rangle &= \sum_{s<t} \gamma_s \eta_s \langle M_s, \lambda \rangle \\
&\leq \frac{2(R_\lambda + 1)}{3} \log \frac{1}{\delta_\lambda} + \sqrt{2 \left( 1 + \sum_{s<t} V_{s,\lambda} \right) \log \frac{1}{\delta_\lambda}} \,,
\end{aligned}
$$

where

$$\delta_\lambda = \frac{3n}{\delta \left( 1 + R_\lambda \right)^2 \left( 1 + \sum_{s<t} V_{s,\lambda} \right)^2} \,.$$

Let $\varepsilon > 0$ and $C > 0$ be constants to be chosen later and define the covering set

$$\Lambda = \{ -C, -C+\varepsilon, \dots, 0, \dots, \varepsilon, \dots, C-\varepsilon, C \}^D \,,$$

which has size $N = |\Lambda| = (2C/\varepsilon)^D$. Then, by the union bound we have with probability at least $1 - \delta$ that

$$\langle S_t, \lambda \rangle \le \frac{2(R_\lambda + 1)}{3} \log \frac{N}{\delta_\lambda} + \sqrt{2 \left(1 + \sum_{s<t} V_{s,\lambda}\right) \log \frac{N}{\delta_\lambda}} \quad \text{for all } \lambda \in \Lambda. \tag{4}$$

From now on, assume this event occurs. Since $F_{t-1}$ does not hold we can apply Lemma 14 to get

$$\left\|G_t^{-1} S_t\right\|_\infty \le \|S_t\|_1 \le 2t^2 D = C.$$

Let $\lambda = G_t^{-1} S_t$ (which is a random quantity) for which $\|\lambda\|_\infty \le C$. Then there exists a $\lambda' \in \Lambda$ such that $\lambda' \le \lambda$ and $\|\lambda' - \lambda\|_\infty \le \varepsilon$.

$$\|S_t\|_{G_t^{-1}}^2 = \langle S_t, \lambda \rangle \le \|S_t\|_1 \varepsilon + \langle S_t, \lambda' \rangle .$$

Therefore

$$\|S_t\|_{G_t^{-1}}^2 \le \|S_t\|_1 \varepsilon + \frac{2(R_\lambda + 1)}{3} \log \frac{N}{\delta_\lambda} + \sqrt{2 \left(1 + \sum_{s<t} V_{s,\lambda}\right) \log \frac{N}{\delta_\lambda}}, \tag{5}$$

where we used the fact that $R_{\lambda_1} \le R_{\lambda_2}$ and $V_{s,\lambda_1} \le V_{s,\lambda_2}$ and $1/\delta_{\lambda_1} \le 1/\delta_{\lambda_2}$ for $\lambda_1 \le \lambda_2$. We now bound the sum term:

$$\sum_{s<t} V_{s,\lambda} \overset{(a)}{\le} \sum_{s<t} \gamma_s \langle M_s, \lambda \rangle^2$$

$$\overset{(b)}{=} \sum_{s<t} \gamma_s (G_t^{-1} S_t)^\top M_s M_s^\top G_t^{-1} S_t$$

$$\overset{(c)}{=} (G_t^{-1} S_t)^\top \sum_{s<t} \gamma_s M_s M_s^\top G_t^{-1} S_t$$

$$\overset{(d)}{\le} S_t^\top G_t^{-1} S_t \overset{(e)}{=} \|S_t\|_{G_t^{-1}}^2, \tag{6}$$

where (a) follows from the definition of $V_{s,\lambda}$ and since if $\nu \in C_s$, then $\gamma_s \le \mathbb{V}[\eta_s | \mathcal{F}_{s-1}]$, (b) by substituting the definition of $\lambda$, (c) by calculation, (d) follows since $\sum_{s<t} \gamma_s M_s M_s^\top < G_t$ and (e) is just the definition. We now set $\varepsilon = 1/(2t^2 D)$ to obtain

$$\|S_t\|_{G_t^{-1}}^2 \overset{(a)}{\le} \|S_t\|_1 \varepsilon + \frac{2(R_\lambda + 1)}{3} \log \frac{N}{\delta_\lambda} + \sqrt{2 \left(1 + \|S_t\|_{G_t^{-1}}^2\right) \log \frac{N}{\delta_\lambda}}$$

$$\overset{(b)}{\le} 1 + \frac{2(R_\lambda + 1)}{3} \log \frac{N}{\delta_\lambda} + \sqrt{2 \left(1 + \|S_t\|_{G_t^{-1}}^2\right) \log \frac{N}{\delta_\lambda}}$$

$$\overset{(c)}{\le} 1 + \frac{2 \left(\frac{2\|S_t\|_{G_t^{-1}}}{\sqrt{\beta}} + 1\right)}{3} \log \frac{N}{\delta_\lambda} + \sqrt{2 \left(1 + \|S_t\|_{G_t^{-1}}^2\right) \log \frac{N}{\delta_\lambda}}, \tag{7}$$

where (a) follows by substituting the previous computation into Eq. (5), (b) since $\|S_t\|_1 \varepsilon \le 1$ by Lemma 14 and the assumption that $F_{t-1}$ does not hold, (c) by Lemma 12 and the assumption that $F_{t-1}$ does not hold. From Eq. (3) and Lemma 12 and the definition of $\beta$ we also obtain

$$\delta_\lambda \le \frac{3n}{\delta \left(1 + \|S_t\|_{G_t^{-1}}^2\right)^2}.$$

By rearranging and naively simplifying Eq. (7), it can be shown that

$$\|S_t\|_{G_t^{-1}} + \sqrt{B} \le 1 + \sqrt{\alpha B} + 2\sqrt{\log \frac{N}{\delta_\lambda}}$$

$$= 1 + \sqrt{\alpha B} + 2\sqrt{\log \left(\frac{3nN \left(1 + \|S_t\|_{G_t^{-1}}^2\right)^2}{\delta}\right)}.$$

The result is finally completed by solving the equation above and choosing

$$\beta = \left(1 + \sqrt{\alpha B} + 2\sqrt{\log\left(\frac{6nN}{\delta}\log\left(\frac{3nN}{\delta}\right)\right)}\right)^2. \qquad \square$$

It remains to prove the lemmas that were used in this proof.

**Lemma 11.** *For any $s < t$, it holds that*

$$\gamma_s \left\| M_s \right\|_{G_t^{-1}} \le \gamma_s \left\| M_s \right\|_{G_s^{-1}}.$$

*Further, if $F_{t-1}$ does not hold, then for all $s < t$ such that $\gamma_s > 4$,*

$$\gamma_s \left\| M_s \right\|_{G_t^{-1}} \le \gamma_s \left\| M_s \right\|_{G_s^{-1}} \le \frac{2}{\sqrt{\beta}} \le 1.$$

*Proof.* The first inequality follows because $G_t \succeq G_s$ and an application of Proposition 8.§2. Since $F_{t-1}$ does not hold, we have $\nu \in C_s$ and since $\gamma_s > 4$ we have from Eq. (2) that one of the following is true:

$$\gamma_s^{-1} \ge \frac{1}{2}\left(\langle M_s, \hat{\nu}_s \rangle + 2\sqrt{\beta}\left\| M_s \right\|_{G_s^{-1}}\right) \ge \sqrt{\beta}\left\| M_s \right\|_{G_s^{-1}}/2;$$

$$\gamma_s^{-1} \ge \frac{1}{2}\left(1 - \langle M_s, \hat{\nu}_s \rangle + 2\sqrt{\beta}\left\| M_s \right\|_{G_s^{-1}}\right) \ge \sqrt{\beta}\left\| M_s \right\|_{G_s^{-1}}/2. \qquad \square$$

**Lemma 12.** *If $F_{t-1}$ does not hold and $\lambda = G_t^{-1}S_t$, then $R_\lambda \le \dfrac{2\left\| S_t \right\|_{G_t^{-1}}}{\sqrt{\beta}}.$*

*Proof.* We apply Lemma 11 to get

$$\gamma_s \langle M_s, \lambda \rangle \overset{(a)}{\le} \frac{2\langle M_s, \lambda \rangle}{\left\| M_s \right\|_{G_t^{-1}}\sqrt{\beta}} \overset{(b)}{=} \frac{2\langle M_s, G_t^{-1}S_t \rangle}{\left\| M_s \right\|_{G_t^{-1}}\sqrt{\beta}} \overset{(c)}{\le} \frac{2\left\| S_t \right\|_{G_t^{-1}}}{\sqrt{\beta}},$$

where (a) follows from Lemma 11, (b) is just the definition of $\lambda$ and (c) is follows from Cauchy-Schwarz. $\qquad \square$

**Lemma 13.** *If $F_t$ does not hold, then $\gamma_t \left\| M_t \right\|_1 \le 4tD$ and $\gamma_t \left\| M_t \right\|_2^2 \le 4tD$.*

*Proof.* The result holds trivially if $\gamma_t = 4$. Suppose $\gamma_t > 4$ and let $\lambda_{\max}$ be the maximum eigenvalue of $G_t$. Then, by Lemma 11, we have

$$\gamma_t \left\| M_t \right\|_2^2 \overset{(a)}{\le} \frac{2}{\sqrt{\beta}}\left\| M_t \right\|_2^2 \left\| M_t \right\|_{G_t^{-1}}^{-1}$$

$$\overset{(b)}{\le} 2\sqrt{D/\beta}\left\| M_t \right\|_2 \left\| M_t \right\|_{G_t^{-1}}^{-1}$$

$$\overset{(c)}{\le} \left\| G_t^{1/2}G_t^{-1/2}M_t \right\|_2 \left\| M_t \right\|_{G_t^{-1}}^{-1}$$

$$\overset{(d)}{\le} \sqrt{\lambda_{\max}},$$

where (a) follows from Lemma 11, (b) by bounding $\left\| M_t \right\|_2 \le \sqrt{D}$, (c) holds by $4D \le \beta$, and (d) follows from Proposition 8.§3. Similarly,

$$\gamma_t \left\| M_t \right\|_1 \le \frac{2}{\sqrt{\beta}}\left\| M_t \right\|_1 \left\| M_t \right\|_{G_t^{-1}}^{-1}$$

$$\le \frac{2\sqrt{D}}{\sqrt{\beta}}\left\| M_t \right\|_2 \left\| M_t \right\|_{G_t^{-1}}^{-1}$$

$$\le \sqrt{\lambda_{\max}}.$$

Now assume that $\gamma_s \|M_s\|_2^2 \leq 4sD$ for all $s < t$, which is immediate if $t = 1$. Then,

$$\gamma_t \|M_t\|_2^2 \leq \sqrt{\lambda_{\max}} \overset{(a)}{\leq} \sqrt{\operatorname{trace}(G_t)}$$
$$= \sqrt{\left(D + \sum_{s=1}^{t-1} \gamma_t \|M_t\|_2^2\right)} \leq \sqrt{\left(D + 4D \sum_{s=1}^{t-1} s\right)}$$
$$= \sqrt{D(1 + 2t(t-1))} \leq 4tD,$$

where (a) again follows from Proposition 8.§3 and the remaining steps are immediate. Therefore by induction, we have $\gamma_t \|M_t\|_2^2 \leq 4tD$ for all $t$. $\qquad\square$

**Lemma 14.** *If $F_{t-1}$ does not hold, then $\|S_t\|_1 \leq 2t^2 D$.*

*Proof.* We use $|\eta_s| \leq 1$ and the previous lemma to get

$$\|S_t\|_1 = \left\| \sum_{s<t} \gamma_s \eta_s M_s \right\|_1 \leq \sum_{s<t} \gamma_s \|M_s\|_1 \leq \sum_{s<t} 4sD \leq 2t^2 D. \qquad\square$$

**Lemma 15.** *If $F_t$ does not hold, then $\gamma_t \min\left\{1, \|M_t\|_{G_t^{-1}}^2\right\} \leq 4 \min\left\{1, \gamma_t \|M_t\|_{G_t^{-1}}^2\right\}$.*

*Proof.* If $\gamma_t = 4$, then the result is trivial. For $\gamma_t > 4$, by Lemma 11, $\gamma_t \|M_t\|_{G_t^{-1}}^2 \leq 1$. Hence, we need to prove $\gamma_t \min\left\{1, \|M_t\|_{G_t^{-1}}^2\right\} \leq 4\gamma_t \|M_t\|_{G_t^{-1}}^2$, which is obvious. $\qquad\square$

# D    Proof of Theorem 1

Define $e_{di} \in \mathbb{R}^{D \times K}$ to be the matrix with $(e_{di})_{ck} = \mathbb{1}\{c = d \text{ and } k = i\}$. For $M \in \mathcal{M}$ we write $\mu(M) = \sum_{k=1}^{K} \psi(\langle M_k, \nu_k \rangle)$ to be the reward for allocation $M$. Given an allocation $M \in \mathcal{M}$ we define the conditional optimal allocation function $M^* : \mathcal{M} \to \mathbb{R}^{D \times K}$ by

$$M^*(M) = \underset{M' \in \mathcal{M}}{\arg\max} \left\{ \sum_{k=1}^{K} \psi(\langle M'_k, \nu \rangle) : M'_{dk} \geq M_{dk} \text{ for all } d \text{ and } k \right\},$$
$$\mu^*(M) = \sum_{k=1}^{K} \psi(\langle M^*(M)_k, \nu_k \rangle),$$
$$\mu^* = \mu^*(0).$$

Note that $M^*(0) = M^*$ is the optimal allocation while $M^*(M)$ is the optimal allocation given that one has committed to allocating at least $M$ already. Let $M_t \in [0,1]^{K \times D}$ be the allocation of Algorithm 1 after $t$ iterations. Assume that $\mu^*(M_{t-1}) = \mu^*$, which is trivial for $t = 1$. Let $(i, d)$ be the task/resource pair selected in the $t$th iteration of Algorithm 1. Suppose that at this point it is sub-optimal to allocate resource $d$ to task $i$. Then

$$\nabla_{e_{di}} \mu^*(M_{t-1}) < 0.$$

This implies that under the optimal allocation, resource $d$ should not be allocated to task $i$ and instead to some other task $j \neq i$. Therefore $\psi(\langle M^*(M_{t-1})_i, \nu_i \rangle) = 1$, since otherwise

$$\nabla_{e_{di} - e_{dj}} \mu^*(M_{t-1}) = \nu_{di} - \nu_{dj} \geq 0,$$

which is a contradiction. Therefore there exists some other resource $1 \leq c \leq D$ that is assigned to task $i$ under the optimal allocation. We choose

$$\alpha = \frac{\nu_{dj} \nu_{ci}}{\nu_{di} \nu_{cj}} \leq 1$$

and compute the derivative, to get

$$\nabla_{e_{di}-e_{ci}\nu_{di}/\nu_{ci}-e_{dj}+e_{cj}\alpha\nu_{di}/\nu_{ci}}\mu^*(M_{t-1})$$
$$= \nu_{cj}\alpha\frac{\nu_{di}}{\nu_{ci}} - \nu_{dj}$$
$$= 0\,,$$

which again implies that allocating resource $d$ to task $i$ is not sub-optimal, which is a contradiction. Therefore $\nabla_{e_{di}}\mu^*(M_{t-1}) = 0$ and so Algorithm 1 is optimal by induction.

## E  Resource Allocation when $\|\nu_k\|_1 \leq 1$

Throughout this subsection we assume that $\|\nu_k\|_1 \leq 1$ for all $k$. Therefore $\psi(\langle M_k, \nu_k\rangle) = \langle M_k, \nu_k\rangle$ for all $M \in \mathcal{M}$ and $k$. Therefore the optimal strategy assigns all of resource $d$ to the task $k$ for which $\nu_{kd}$ is the greatest:

$$M^*_{kd} = \mathbb{1}\left\{k = \arg\max_i \nu_{id}\right\}\,,$$

where ties are broken arbitrarily. The algorithm operates by maintaining a set of tasks for each resource that are plausibly still optimal. Each resource type is then allocated to a single task in this set uniformly at random, with tasks being removed from this set in phases as the algorithm proves that allocating a particular resource to this task is sub-optimal with high probability. The structure of the problem then allows us to simultaneously estimate all parameters of $\nu$ using importance sampling, which ultimately leads to an optimal rate. The algorithm is easily implemented to run in $O(KD)$ per iteration.

---

**Algorithm 3** Unconstrained Allocation Algorithm

---

1: **Input:** $K$, $D$, $n$, $\delta$
2: $\mathcal{A}_d := [K]$ and $\Delta_d := 1$ and $\tau_d := n_d := 0$
3: **for** $t \in 1, \ldots, n$ **do**
4:     **for** $d \in 1, \ldots, D$ **do**
5:         **if** $t = \tau_d + 1$ **then**
6:             $\hat{\nu}_{kd} := \frac{1}{n_d}\sum_{s=\tau_d-n_d+1}^{\tau_d} Z_{skd}Y_{sk}$
7:             $\mathcal{A}_d := \mathcal{A}_d \cap \{k : \hat{\mu}_{kd} + 2\Delta_d \geq \max_j \hat{\mu}_{jd}\}$
8:             $\Delta_d := \Delta_d/2$
9:             $n_d := n(|\mathcal{A}_d|, \Delta_d)$ and $\tau_d := \tau_d + n_d$
10:         **end if**
11:         $I_{td} \sim \text{Uniform}(\mathcal{A}_d)$
12:         Choose $M_{tkd} := \mathbb{1}\{k = I_{td}\}$
13:         $Z_{tkd} := |\mathcal{A}_d|M_{tkd} - \frac{|\mathcal{A}_d|}{|\mathcal{A}_d|-1}(1 - M_{tkd})$
14:     **end for**
15:     Observe reward $Y_{tk} \sim \text{Bernoulli}(\langle M_{tk}, \nu_k\rangle)$
16: **end for**
17: **function** $n(m, \Delta)$
18:     **Return** $\left\lceil\frac{2(6 + 3m + \Delta)}{3\Delta^2}\log\frac{2}{\delta}\right\rceil$
19: **end function**

---

## F  Regret of Algorithm 3

**Theorem 16.** *Define $\nu^*_d = \max_k \nu_{kd}$ and $\Delta_{kd} = \nu^*_d - \nu_{kd} \geq 0$. The regret of Algorithm 3 when run with $\delta = (DKn)^{-2}$ is at most*

$$R_n \in O\left(\sum_{d=1}^D \sum_{k:\Delta_{kd}>0} \frac{\log nKD}{\Delta_{kd}}\right)\,. \tag{8}$$

**Corollary 17.** *The regret of Algorithm 3 satisfies $R_n \in \tilde{O}(D\sqrt{Kn})$ in the worst-case.*

The proof of the corollary is omitted, but follows from standard arguments for converting from problem-dependent to problem-independent regret bounds (Bubeck and Cesa-Bianchi [2012] and others).

Before presenting the analysis we compare the regret bound of Theorem 16 to the well-known problem dependent bounds for finite-armed bandits, which look the same as Eq. (8), but with $D = 1$. An incautious reader might believe that a bound similar to Eq. (8) could be derived by simultaneously running $D$ copies of some optimal bandit algorithm. But this is not the case because the algorithm observes a reward for each task and not for each resource. Alternatively one could ignore the semi-bandit feedback and apply an algorithm designed for stochastic linear bandits. This approach also leads to sub-optimal bounds because the $K$ will appear outside of the square root. The optimal regret can only be obtained be exploiting the special structure of the problem. Notably, that if only a small amount of resources are allocated to a particular task, then the probability that it is completed is close to zero and hence the variance of the outcome is significantly reduced. This low variance can then be exploited to accelerate the rate of estimation of the parameters beyond what is normally possible.

*Proof of Theorem 16.* We will analyse the regret using the following decomposition

$$R_n = \sum_{t=1}^{n}\sum_{d=1}^{D}\nu_d^* - \mathbb{E}\left[\sum_{t=1}^{n}\sum_{k=1}^{K}\langle M_{tk}, \nu_k\rangle\right] = \sum_{d=1}^{D}\mathbb{E}\left[\sum_{t=1}^{n}\Delta_{dI_{td}}\right].$$

Now we fix $d$ and analyse the expectation inside the sum. Let $\tau_{d1}, \tau_{d2}, \ldots$ be the sequence of values of $\tau_d$ as it is updated in Line 9 of the algorithm, and let $n_{d1}, n_{d2}, \ldots$ be the corresponding sequence of the values of $n_d$. Similarly, let $\mathcal{A}_{d1}, \mathcal{A}_{d2}, \ldots$ be the sequence of sets of active tasks.

$$\mathbb{E}\left[\sum_{t=1}^{n}\Delta_{dI_{td}}\right] = \mathbb{E}\left[\sum_{\ell=1}^{\infty}\sum_{t=\tau_{d\ell}-n_{d\ell}+1}^{\tau_{d\ell}}\Delta_{dI_{td}}\right] = \sum_{\ell=1}^{\infty}\mathbb{E}\left[\sum_{t=\tau_{d\ell}-n_{d\ell}+1}^{\tau_{d\ell}}\Delta_{dI_{td}}\right]$$

$$= \sum_{\ell=1}^{\infty}\mathbb{E}\left[n_{d\ell}\sum_{k\in\mathcal{A}_{d\ell}}\frac{\Delta_{kd}}{|\mathcal{A}_{d\ell}|}\right] \leq \sum_{\ell=1}^{\infty}\mathbb{E}\left[\frac{20}{3(2^{-\ell})^2}\left(\log\frac{2}{\delta}\right)\sum_{k\in\mathcal{A}_{d\ell}}\Delta_{kd}\right]. \quad (9)$$

Shortly we are going to show with sufficiently high probability that for $\ell \geq \lceil -\log_2(\Delta_{kd}/4)\rceil$ we have $k \notin \mathcal{A}_{d\ell}$ and that $k^*d = \arg\max_k \nu_{kd}$ is in $\mathcal{A}_{d\ell}$. Therefore

$$(9) \leq \sum_{k=1}^{K}\sum_{\ell=1}^{\lceil -\log_2(\Delta_{kd}/4)\rceil}\frac{20\Delta_{kd}}{3(2^{-\ell})^2}\log\frac{2}{\delta} \leq \sum_{k=1}^{K}\frac{16\cdot 20\cdot 4}{3\Delta_{kd}}\log\frac{2}{\delta}.$$

Let $\{\mathcal{F}_t\}_{t=1}^n$ be the filtration of information available up to each time step. Let $\tau_{d\ell} - n_{d\ell} + 1 \leq t \leq \tau_{d\ell}$. A straightforward computation shows the following results:

1. $\mathbb{E}[Z_{tkd}Y_{tk}|\mathcal{F}_{t-1}] = \nu_{kd}$.
2. $Z_{tkd}Y_{tk} \in \{0, |\mathcal{A}_d|\}$.
3. $\mathbb{V}[Z_{tkd}Y_{tk}|\mathcal{F}_{t-1}] \leq |\mathcal{A}_d| + 2$.

Let $\hat{\nu}_{kd}$ be the estimate of $\nu_{kd}$ made at time step $\tau_\ell$ in Line 6 of Algorithm 3. We apply the martingale version of Bernstein's inequality (Theorem 3.15 by McDiarmid [1998]) to obtain

$$\mathbb{P}\left\{|\hat{\nu}_{kd} - \nu_{kd}| \geq 2^{-\ell}\right\} = \mathbb{P}\left\{\left|\sum_{s=\tau_d-n_d+1}^{\tau_d}Z_{skd}R_{sk} - n_d\nu_{kd}\right| \geq n_d 2^{-\ell}\right\}$$

$$\leq 2\exp\left(-\frac{n_d(2^{-\ell})^2}{2\left(|\mathcal{A}_d| + 2 + \frac{|\mathcal{A}_d|2^{-\ell}}{3}\right)}\right) \leq \delta.$$

But if $|\hat{\nu}_{kd} - \nu_{kd}| \leq 2^{-\ell}$ for all $k$, then (a) $k_d^*$ is not removed from $|\mathcal{A}_d|$ and (b) $\ell \geq \lceil -\log_2(\Delta_{kd}/4)\rceil$ implies that $k$ is removed from $\mathcal{A}_d$. The probability that there exists a resource $d$, phase $\ell$ and $k$ such that $|\hat{\nu}_{kd} - \nu_{kd}| \geq 2^{-\ell}$ in the $\ell$th phase is bounded using the union bound by $DK\ell\delta \leq DKn\delta$.

Therefore if Algorithm 3 is run with $\delta = (DKn)^{-2}$, then the contribution of the regret to the failure of any confidence set is at most 1, which leads to a regret bound

$$R_n \le 1 + \frac{4 \cdot 16 \cdot 20}{3} \sum_{d=1}^{D} \sum_{k=1}^{K} \frac{1}{\Delta_{kd}} \log \frac{2}{\delta}$$

as required. $\qquad\square$

## G  Linear Bandits on the Hypercube

The importance sampling approach used in the previous section can be applied to linear stochastic bandits on the hypercube. In this case the algorithm chooses $M_t \in [0,1]^d$ at each time step and receives reward $Y_t = \langle M_t, \nu \rangle + \eta_t$ where $\nu \in \mathbb{R}^d$ satisfies $\|\nu\|_1 \le 1$ and $\eta_t \in [-1,1]$ has zero mean. Note that $\nu$ may be negative in some dimensions, so the optimal strategy is not knowable in advance.

---

**Algorithm 4**

---

1: **for** $t \in 1, \ldots, n$ **do**
2:     **for** $d \in 1, \ldots, D$ **do**
3:         $\hat{\nu}_{td} = \dfrac{\sum_{\tau=1}^{t-1} \psi_{td} M_{td} Y_t}{\sum_{\tau=1}^{t-1} \psi_{td}}$
4:         $c_{td} = \sqrt{\dfrac{2 \log(2n^2)}{\sum_{\tau=1}^{t-1} \psi_{td}}}$
5:         Sample $X_{td} \in \{-1, 1\}$ with $\mathbb{P}\{X_{td} = 1\} = 1/2$
6:         $\psi_{td} = \begin{cases} 1 & \text{if } \hat{\nu}_{td} \in (-c_{td}, c_{td}) \\ 0 & \text{otherwise} \end{cases}$
7:         $M_{td} = \begin{cases} 1 & \text{if } \hat{\nu}_{td} - c_{td} > 0 \\ -1 & \text{if } \hat{\nu}_{td} + c_{td} < 0 \\ X_{td} & \text{otherwise} \end{cases}$
8:     **end for**
9: **end for**

---

**Theorem 18.** *The regret of Algorithm 4 is at most* $R_n \le 3 \|\nu\|_1 + \displaystyle\sum_{d:\nu_d \neq 0} \frac{2 \log(2n^2)}{|\nu_d|}$.

This result is especially nice because (a) the algorithm is efficient, (b) the bound scales optimally with the dimension, (c) the problem-dependent bound is essentially correct, and finally (d), the bound is adaptive to sparsity in $\nu$ with no dependence on $D$ if $\nu$ is sparse. The algorithm and proof are significantly more straight-forward than above as the Bernstein's inequality and phases can be replaced by straight-forward union bounds in combination with Azuma's inequality. Details may be found in Appendix G.

*Proof of Theorem 18.* Let $\{\mathcal{F}_t\}$ be the filtration with $\mathcal{F}_t$ containing information up to time step $t$. Then $\hat{\nu}_{td}$, $c_{td}$ and $\psi_{td}$ are all $\mathcal{F}_{t-1}$-measurable. First we note that $Y_t \in [-2, 2]$ and that if $\psi_{td} = 1$, then $\mathbb{E}[\psi_{td} M_{td} Y_t | \mathcal{F}_{t-1}] = \nu_d$ and $\psi_{td} M_{td} Y_t \in [-2, 2]$. Let $F_d$ be the event that there exists a $t \le n$ for which $|\hat{\nu}_{td} - \nu_d| > c_{td}$. By Azuma's inequality and the union bound we have $\mathbb{P}\{F_d\} \le 1/n$.

We now decompose the regret

$$R_n = \mathbb{E}\left[\sum_{t=1}^{n}\sum_{d=1}^{D}(|\nu_d| - M_{td}\nu_d)\right]$$

$$= \sum_{d=1}^{D}\mathbb{E}\left[\mathbb{1}\{F_d\}\cdot 2n|\nu_d| + \mathbb{1}\{\neg F_d\}\sum_{t=1}^{n}(|\nu_d| - M_{td}\nu_d)\right]$$

$$\leq 2\|\nu\|_1 + \sum_{d=1}^{D}\mathbb{E}\left[\mathbb{1}\{\neg F_d\}\sum_{t=1}^{n}(|\nu_d| - M_{td}\nu_d)\right]$$

$$= 2\|\nu\|_1 + \sum_{d:\nu_d\neq 0}|\nu_d|\left\lceil\frac{2\log(2n^2)}{|\nu_d|^2}\right\rceil$$

$$\leq 3\|\nu\|_1 + \sum_{d:\nu_d\neq 0}\frac{2\log(2n^2)}{|\nu_d|}.$$

The second last line follows from two facts. First, if $\psi_{td} = 0$ and $\neg F$, then $|\nu_d| - M_{td}\nu_d = 0$. Second, if $\neg F$ and

$$\sum_{\tau=1}^{t-1}\psi_{\tau d} > \left\lceil\frac{2\log(2n^2)}{|\nu_d|^2}\right\rceil,$$

then $\psi_{td} = 0$. $\qquad\square$

**Remark 19.** With only a little effort this algorithm could be made anytime. It may also be possible to make the (already quite small) constants smaller.