[Reviews · NeurIPS 2015]

Submitted by Assigned_Reviewer_1

The paper presents an interesting extension of stochastic linear MAB problems. The main difference with the latter is that the average reward is linear up to point after which it is clipped. The difference does not seem very important but certainly yields technicalities. The authors do not provide much examples of applications, and this might be a concern about this kind of problems -- there have been a lot of problems with similar flavour in the literature, and yet no real applications as far as I am aware.

The proposed algorithm (Optimistic Allocation Algorithm) is present in Algorithm 2, without much intuitive explanations, or connections with algorithms previously proposed for similar problems. The reader would appreciate such explanations and comparisons to earlier algorithms to be able to judge the novelty of the approach.

The regret of the algorithm is analysed in Section 4. The analysis is sound, and well presented. It would be also nice here to know what is the major novelty here compared to existing regret analysis techniques.
Summary: The paper presents a class of online stochastic optimization problems that extends linear MAB problems with semi-bandit feedback, and resource allocation problems. The paper is well written and structured. The results are interesting. The techniques used to establish regret upper bounds are however extensions of existing methods.

Submitted by Assigned_Reviewer_2

The paper is clear, even if some part of the analysis of section 4 could have been simplified. The proposed algorithm seems original to the best of my knowledge and the simplification of the allocation problem that allows for formal regret analysis does not seem to penalize the potential impact and application of the proposed optimistic allocation algorithm of the paper. The robustness to noise could have been a plus to the experimental section even if the experiment sounds quite convincing.
Summary: The paper is pretty well written and tackle in a quite novel way the classic problem of resource allocation and proposed an optimistic in face of the uncertainty principle type of algorithm to solve the problem in a simplified linearized setting. Thanks to this relaxation, the author provides regret analysis of the proposed allocation algorithm.

Submitted by Assigned_Reviewer_3

Summary : The paper proposes a new setting for addressing resource allocation problems, which comes along with an algorithm designed for this problem. Theoretical guarantees on its performance in terms of the regret are provided.

Quality / Clarity : the paper is very well written and organized. In particular, there is a real effort to make it pedagogical.

Originality : the proposed setting is new, as well as the proposed algorithm.

Additional information : the present reviewer has not gone through the proofs.
Summary: A very strong paper about resource allocation using a bandit setting.

Submitted by Assigned_Reviewer_4

The choice of this reward clipping could be discussed more. Why not, for instance considering using a sigmoid function. It would then have connections with

generalised linear model and especially the work of Filippi et al 2010. ?

In Experiments: - If I understand well, in Fig. 2 the two weights are the weights of the weighted versions of the algorithm. But the fact that you use the same color (blue/red) and shapes (square/circle) that in Figure 1. makes it misleading. - At what time t is computed the regret in Fig.3? How would time influence this graph?
Summary: The paper addresses a problem of parallel allocation of resources in order to solve multiple task at the same time. The proposed approach smartly makes use of the constraint and of the noise model to improve upon direct application of linear bandit algorithm in this case.

Submitted by Assigned_Reviewer_5

In this paper, the authors extend the "resource allocation with semi-bandit feedback", proposed by Lattimore et al. [2014], to the multi-resource case. The paper has provided two regret bounds, one for the worst case (Theorem 2) and the other for the "resource-laden" case (Theorem 7). The authors also provide a new result on the "weighted least squares estimation", which is independently interesting.

The paper is well-written and very interesting, the analysis in this paper is also rigorous. The extension to the multi-resource case is non-trivial, and the new result on the "weighted least squares estimation" is very interesting and might be reused by researchers in the field of bandit/RL in the future. Thus, I think this paper meets the acceptance threshold.

My main concern about this paper is its practical performance. As the authors have pointed out, the optimization problem in Line 8 of Algorithm 2 is likely to be NP-hard, and some approximation algorithms (e.g. "alternative optimization") must be used in practice. It is not clear to me that when such approximation algorithms are used, whether or not the performance of the proposed algorithm (measured in regret) is still good. Specifically, in Section 6, the authors only report experiment results in two baby examples (both with D=K=2), which are not convincing.

Two minor comments:

1) In this paper, the authors assume that all the resource types have budget 1. It is not clear to me if this "uniform budget assumption" is essential for the results of this paper. Please explain!

2) In Line 39-40, the authors assume that the kth row is associated with the kth task, while in Line 361-362, the authors assume that the kth column is associated with the kth task. Please modify the paper to make them consistent.

************************************************************************************************************************* The authors' rebuttal has addressed my minor comments. However, it has not addressed my main concern. So I think 6 is an appropriate score.
Summary: This paper is interesting, well-written, and rigorous, and I think it meets the acceptance threshold. However, the experiment results of this paper are not convincing, and it is not clear to me if the proposed algorithm will work well in practice.

Author Feedback
Author rebuttal: =====================================
General response to all reviewers.
=====================================

Thank you to all reviewers for your careful reading and helpful suggestions. Please find specific responses below.

Many reviewers are concerned about that the main algorithm is not computationally efficient. While we are also concerned about this, we think that establishing basic results such as in our paper are important on their own that further research can build on, as it happened in the past for other similar settings.

=====================================
Reviewer 1.
=====================================

We would point out two other differences between this setting and linear bandits. First, the noise is heteroscedastic, and this must be exploited to get good performance. Second, the feedback is semi-bandit and the action-sets have a complicated structure due to the resource constraints.

The analysis is indeed very reminiscent of the linear bandit analysis, but care is taken to exploit the noise model in an optimal way. In particular, by carefully applying Theorem 3, which to our knowledge is the first of its type for weighted least squares estimators.

We'll add further text to the paper to emphasize these points in addition to what is currently present in the introduction.

=====================================
Reviewer 3.
=====================================

We agree that the practical performance of the algorithm deserves more attention, and particularly how best to approach the optimisation problem. As we remark in the conclusion, the most natural direction is to try and generalise Thompson sampling, which has good empirical performance for linear bandits. Since this is highly non-trivial we leave this for future work.

Thank you for spotting the error with the matrix orientation.

Regarding the uniform bound on the resource constraints. No, it is not necessary, but it is also non-restrictive, since you can scale the units of the resource so that the limit is 1 and the corresponding $\nu$ is scaled. We will add a remark to the final version if accepted.

=====================================
Reviewer 4.
=====================================

Thank you for your review.

=====================================
Reviewer 5.
=====================================

The sigmoid function would have been another natural choice, but to our eyes it would have complicated an already tricky situation and we felt it better to focus on one difficulty at a time, which here is how to exploit the noise model in an optimal way. We expect many of the techniques used here could be combined
with the techniques used by Filippi et. al. to obtain a comparable result. We will consider this more carefully before a final version and add a remark if true.

Thank you for the comments on the experiments. We will change Fig. 2 to make this distinction clear. In Fig. 3 the horizon was set to 500,000 (we mention this on Line 369, but will add to the figure). For different horizons we would not expect a significant difference in the shape of the graph, except that it will shift and expand/shrink in a similar way as gap-dependent fixed horizon regret plots for bandits do.

=====================================
Reviewer 6.
=====================================

We agree the main drawback of this work is the inefficiency of the main algorithm. While the problem formulation is a straight-forward extension of Lattimore et. al. 2014, this extension has practical significance (see the example in the introduction), while it also presented significant new challenges both in the algorithm design and analysis (both the concentration inequalities and the analysis are quite different, with the latter having more in common with linear bandits). That a relatively small change to the problem setting causes such notable differences might be instructive on its own. Of course it would be nice to include more content in the main body, especially
the efficient algorithm, but a (primarily) theory paper with all proofs in the appendix is also not optimal, and the efficient algorithm has a very different flavour to the inefficient one that would require us to introduce more notation and explanations of the approach.

=====================================
Reviewer 7.
=====================================

Thank you for your review. It's not clear to us precisely what is meant by "robustness to noise". Perhaps that the algorithm will still work
if the assumptions are not quite met? Or if the returns are occasionally perturbed in some non-iid way? The algorithm is quite conservative, so
we expect it will perform acceptably in this situation.